# REPLACING MULTI-LAYER PERCEPTRON WITH TAYLOR EXPANSION

## ABSTRACT

The Multi-Layer Perceptron (MLP) constitutes a fundamental building block of deep learning. However, its inherent reliance on dense matrix multiplication imposes a substantial computational burden, especially on resource-constrained edge devices, which poses a critical challenge to the real-time inference requirements of edge artificial intelligence applications. In this paper, we propose T-MLP, a novel method that mitigates this challenge by approximating the MLP via a lightweight module that combines K-means and first-order Taylor expansion. Replaced by T-MLP, MLP online inference is reduced to a single K-means prediction, followed by one dense matrix multiplication and two element-wise additions. On a commercial edge device, T-MLP yields $>$**100×** speed-up on large-scale MLP with $< 1\%$ accuracy loss. Grounded in reduced time-complexity and hardware-friendly footprint, T-MLP establishes a new paradigm for edge-side efficient inference.

## 1 INTRODUCTION

The Multi-Layer Perceptron (MLP)(Popescu et al., 2009) is the foundational substrate of modern deep learning. An input layer, a output layer, and multiple hidden layers with nonlinear activation functions endow the MLP with rich representational power for complex classification and regression. With the rise of edge artificial intelligence, MLPs are ubiquitously deployed on edge devices across a diverse range of tasks, including image classification(Lin et al., 2020; 2021; Howard et al., 2017; Sandler et al., 2018), network intrusion detection(Yin et al., 2023; Rosay et al., 2019; de Almeida Florencio et al., 2018), traffic monitoring(Pan et al., 2021; Sun et al., 2025; Jabakumar, 2023; Zhang et al., 2024), automatic speech recognition(Malik et al., 2021; Siniscalchi et al., 2014) and various AIoT-enabled deployment scenarios(Chi et al., 2024; Fatan et al., 2016; Miryala et al., 2022; Sivapalan et al., 2022). Yet, edge devices yield $< 1\%$ of cloud servers' peak FLOPS, especially on edge devices without GPU or NPU. The dense matrix multiplication inherent to MLPs imposes a significant computational burden on resource-constrained edge devices, which severely compromises the real-time inference deadlines of edge AI applications, as Figure 1 demonstrates.

Despite the pressing demand for real-time MLP inference at the edge, a general and effective solution remains absent. Existing optimization methods such as quantization and pruning primarily aim at compression; inference acceleration is a limited by-product, which typically yield $< 4\times$ speed-up(Xiao et al., 2023; Liu et al., 2025). In this paper, we introduce a simple yet effective solution: replacing the MLP with first-order Taylor expansion. Our insight is straightforward that the first-order Taylor expansion can transform an MLP, which involves multiple matrix multiplications, into a single matrix multiplication, provided that the rele-

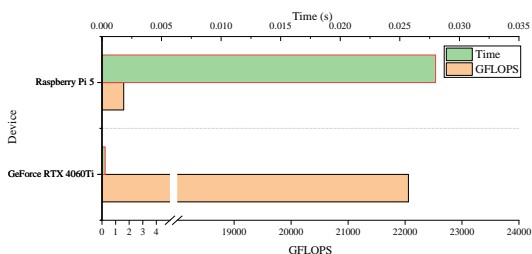

Figure 1: **Edge vs. Commercial GPU:** computational power comparison and latency of large scale matrix multiplication

vant coefficients of the Taylor expansion are precomputed. Due to the nature of Taylor expansion, the approximation accuracy is contingent on the distance between the expansion point and the actual input. Thus, to confine the bulk of the input distribution to the high-probability vicinity of

each expansion point, we perform K-means clustering, adopt the resulting centroids as expansion points, and precompute and cache the Taylor coefficients at each centroid. In this way, we transform the MLP to a single K-means prediction, followed by one dense matrix multiplication, dramatically speeding up inference while preserving the original prediction accuracy. Experimental results demonstrate that out method achieves $>$**100×** speed-up for large-scale MLPs on edge devices.

We provide theoretical complexity analysis demonstrating the superiority of our method over the native MLP, and empirically validate it on commercial edge hardware. Further experiment explored the impact of width, height, and activation choice of MLP on the approximation fidelity. Finally, we analyze the conditions under which optimal efficacy is attained.

In summary, our contributions are as follows:

1) To the best of our knowledge, we are the first to leverage approximation for MLP acceleration, offering a novel paradigm for efficient edge inference.

2) We derive a rigorous error upper bound for our method and systematically analyze how hardware characteristics influence its efficacy.

3) We conduct experiments on commercial off-the-shelf edge devices and validate the practical efficacy of our method. And experimental results demonstrate that our method achieves $>$**100×** speed-up for large-scale MLPs on edge devices.

## 2 RELATED WORK

### 2.1 MLP OPTIMIZATION METHODS

To enable efficient MLP inference, existing studies predominantly optimize MLPs via model compression. Common compression techniques include quantization, pruning and Low-rank Factorization, etc(Dantas et al., 2024). Quantization is a method that converts high-bit floating-point parameters to low-bit integers, enabling efficient model inference(Gholami et al., 2022; Kim et al., 2023). Leveraging low-bit computation, quantization enables acceleration on general-purpose CPUs; however, the speed-up is inherently hardware-constrained and typically capped at $< 4\times$(Xiao et al., 2023; Frantar et al., 2022). Pruning accelerates inference by structurally or non-structurally eliminating unimportant weights or neurons, thereby reducing model redundancy(Vadera & Ameen, 2022). However, the sparsity introduced by pruning yields marginal acceleration, typically less than $2\times$(Hu & Yuan, 2025; Liu et al., 2025). Low-rank factorization accelerates inference by approximating the original large matrix with the product of two smaller matrices(Goyal et al., 2019; Ou et al., 2024). Yet, the actual acceleration is highly dependent on the factorization quality and hardware mapping, thus speed-up is not guaranteed.

In essence, these techniques are devised for model compression; inference acceleration is not their primary objective. Besides, our approach is orthogonal to existing methods and can be concatenated to yield simultaneously compact and highly efficient models for edge devices.

### 2.2 APPROXIMATE MATRIX MULTIPLICATION

For acceleration-oriented inference, approximation serves as a potential viable paradigm. Approximate Matrix Multiplication (AMM) constitutes a pivotal research avenue for accelerating large-scale matrix products. The prevailing paradigm projects the operand matrices into a lower-dimensional space and subsequently performs an exact multiplication on the compressed representations(Liberty, 2013; Ghashami et al., 2016; Teng & Chu, 2019). Representative strategies include matrix sketching algorithms that deterministically(Francis & Raimond, 2022; Huang, 2019; Luo et al., 2019) or randomly construct projection matrices to curtail computational cost(Nelson & Nguyên, 2013; Dasgupta et al., 2010; Pagh, 2013; Kyrillidis et al., 2014; Sarlos, 2006). These methods only consider each matrix in isolation. And there are works that introduce variations which take into account both matrices(Francis & Raimond, 2018; Ye et al., 2016; Mroueh et al., 2016).

More recently, several works(Blalock & Guttag, 2017; 2021) have borrowed the insight of Product Quantization (PQ)(Ge et al., 2014): the vector product $A^T B$ is transformed into a sequence of table lookups and several additions. Concretely, the input vector A is partitioned into disjoint sub-vectors $a_i$; each $a_i$ is assigned to its nearest centroid $c_i$ obtained by clustering over the training data. The partial inner products $c_i^T b_i$ are precomputed and stored in a lookup table, so the online evaluation of $A^T B$ reduces to centroid indexing, table retrieval, and several additions, drastically shrinking the arithmetic complexity.

Originally devised for isolated matrix multiplications, these approaches cannot approximate a complete MLP; yet, the product-quantization philosophy inspires us to approximate the MLP directly. Instead of approximating isolated matrix products via discrete lookup tables, we approximate the entire MLP as a continuous, piecewise-linear function.

## 3 METHOD

To enable real-time MLP inference on edge devices, we propose an approximation-centric acceleration scheme. We first presents our method, termed T-MLP, and then provides a theoretical justification of its efficacy.

### 3.1 TAYLOR EXPANSION COUNTERPART

Our method approximates an MLP via first-order Taylor expansion:

$$MLP(x) \approx MLP(c_i) + \nabla MLP(c_i)^T (x - c_i). \tag{1}$$

which comprises two distinct phases: the offline phase (Steps 1–3) for precomputing $MLP(c_i)$ and $\nabla MLP(c_i)^T$ and the inference phase (Step 4) for dispatching and calculating, as illustrated in Figure 2.

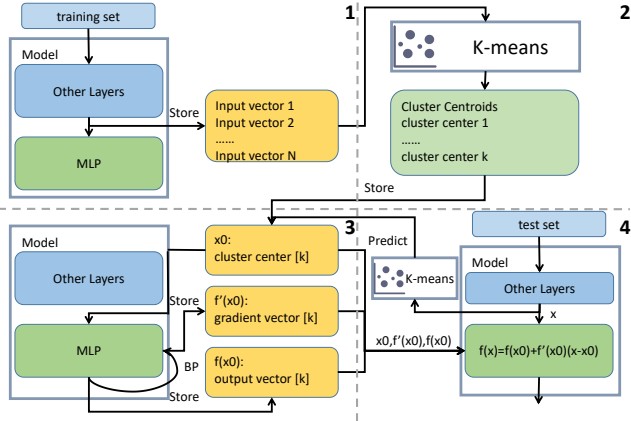

Figure 2: Flow chart of T-MLP. **Step 1:** collect the input fed to MLP. **Step 2:** cluster the input. **Step 3:** precompute and store the Taylor expansion coefficients. **Step 4:** inference with T-MLP.

During the offline phase, we first identify the Taylor expansion points and precompute the corresponding outputs and gradients. The fidelity of the first-order Taylor approximation degrades with the Euclidean distance $|x - c_i|$. To minimize this discrepancy across the entire data manifold, we apply K-means clustering on the training inputs to obtain a representative set of centroids $\{c_i\}_{i=1}^k$ and perform Taylor expansion about these centroids. To identify the centroids, we first perform the forward pass of the well trained model over the training data set and collect every input vectors fed to the MLP, which are subsequently compressed into $k$ representative centroids by K-means. For every centroid $c_i$, we perform the forward pass again to obtain output as $MLP(c_i)$, and then perform back-propagation to compute the corresponding gradient $\nabla MLP(c_i)^T$.

During the inference phase, an incoming input $x$ is dispatched to the nearest centroid $c_i$ in the learned code-book. And the final prediction is then delivered by the first-order Taylor expansion centred at $c_i$.

## 3.2 THEORETICAL ANALYSIS

A typical MLP can be regarded as a highly intricate composite function:

$$f(x) = W_n(\sigma(\ldots \sigma(W_1(x)))), \tag{2}$$

Applying a first-order Taylor expansion to this complex composite function yields an elegant linear approximation (T-MLP):

$$f(x) \approx f(x_0) + \nabla f(x_0)^T (x - x_0), \tag{3}$$

where $f(x_0)$ and $\nabla f(x_0)$ represents the MLP's output and first-order gradient at the expansion point $x_0$, which can be obtained and cached offline. With T-MLP, the MLP is transformed to a single K-means prediction, followed by one dense matrix multiplication and two element-wise additions.

Considering the time complexity, a conventional MLP requires $O(N_1 \times N_2 + \cdots + N_{L-1} \times N_L)$ operations, where $L$ represents the number of layers of MLP, and $N_i$ represents the vector dimension. Each term denotes the time complexity of one linear layer. And the time complexity of the T-MLP is $O(N_1 \times N_L + k \times N_1)$, where $k$ is the number of cluster centers in K-means model. Evidently, for a deep and wide MLP, provided the k is set to an appropriate magnitude, T-MLP is able to effectively reduces the time complexity.

Meanwhile, by exploiting the integral-form remainder of the Taylor expansion, we establish an upper bound on the approximation error between our proposed method and the original MLP:

**Theorem 3.1** (T-MLP upper error bound). *Let $f : \mathbb{R}^d \to \mathbb{R}$ be twice continuously differentiable on the line segment joining $c_i$ and $x$. Then the approximation error satisfies*

$$\|f(x) - \hat{f}(x)\| \leq \frac{1}{2} \sup_{z \in [c_i, x]} \|\nabla^2 f(z)\|_{\text{op}} \cdot \|x - c_i\|^2. \tag{4}$$

The detailed derivation is provided in the Appendix.

Theoretical analysis establishes that computational savings are guaranteed whenever the asymptotic cost of the Taylor expansion-based approximation falls below that of the standard MLP. In theory, a larger $k$ yields higher approximation accuracy at the expense of diminishing speed-up; however, the practical trade-off is more nuanced, and our experiments demonstrate that $k$ exerts only a marginal impact on the overhead of the proposed method.

## 4 EXPERIMENTS

In this section, to empirically validate the proposed approach. To the best of our knowledge, this work presents the first attempt to accelerate MLP inference through approximation; therefore, we adopt the original MLP as our primary baseline. We evaluate its performance on two representative edge workloads: image classification and network-intrusion detection. All experiments are conducted on a Raspberry Pi 5 (4 GB) on a single CPU core, a commodity edge device equipped with a 64-bit Arm Cortex-A76 processor without discrete GPU or NPU.

## 4.1 IMAGE CLASSIFICATION

Image classification is the gateway task for vision applications on edge devices. For the image-classification task, we trained the classical VGG-16(Simonyan & Zisserman, 2014) model on the CIFAR-100 dataset(Krizhevsky, 2009). The model stacks thirteen convolutional layers and terminates in a three-stage MLP classifier whose fully-connected layers are dimensioned as (512, 4096), (4096, 4096), and (4096, 100), each followed by ReLU activations. After 200 epochs of training, the model achieves 72.2 % top-1 accuracy on the CIFAR-100 test set. We replace the classifier of VGG16 with T-MLP, with the number of centroids progressively increasing from 5 to 320. The experiment results are depicted in Tables 1 and Figures 3.

Table 1: VGG 16 Experiments on Raspberry Pi 5. ACC denote the accuracy of T-MLP. $T_{conv}$, $T_{MLP}$, $T_{kmeans}$, $T_{expansion}$ respectively denote the latency of the convolutional layers, the original MLP, the K-means prediction, and the Taylor expansion computation. $T_{conv+MLP}$ and $T_{conv+T-MLP}$ reports the latency of original VGG16 and our proposed method.

| K | Acc | MSE | $T_{conv}$ | $T_{MLP}$ | $T_{kmeans}$ | $T_{expansion}$ | $T_{conv+MLP}$ | $T_{conv+T-MLP}$ |
|---|-----|-----|------------|-----------|--------------|-----------------|----------------|------------------|
| 5 | 72.09 | 156.04 | 0.07448 | 0.30363 | 1.05E-4 | 1.51E-4 | 0.37811 | 0.06878 |
| 25 | 72.21 | 60.43 | 0.07304 | 0.31335 | 1.17E-4 | 1.54E-4 | 0.3864 | 0.06950 |
| 60 | 72.03 | 28.19 | 0.0649 | 0.30377 | 1.45E-4 | 1.56E-4 | 0.36866 | 0.06636 |
| 100 | 72.17 | 13.48 | 0.07684 | 0.31489 | 1.68E-4 | 1.57E-4 | 0.39173 | 0.07061 |
| 180 | 72.35 | 11.87 | 0.06977 | 0.31583 | 2.20E-4 | 1.63E-4 | 0.38447 | 0.06626 |
| 320 | 72.17 | 10.61 | 0.06977 | 0.30252 | 3.29E-4 | 1.64E-4 | 0.37114 | 0.06599 |

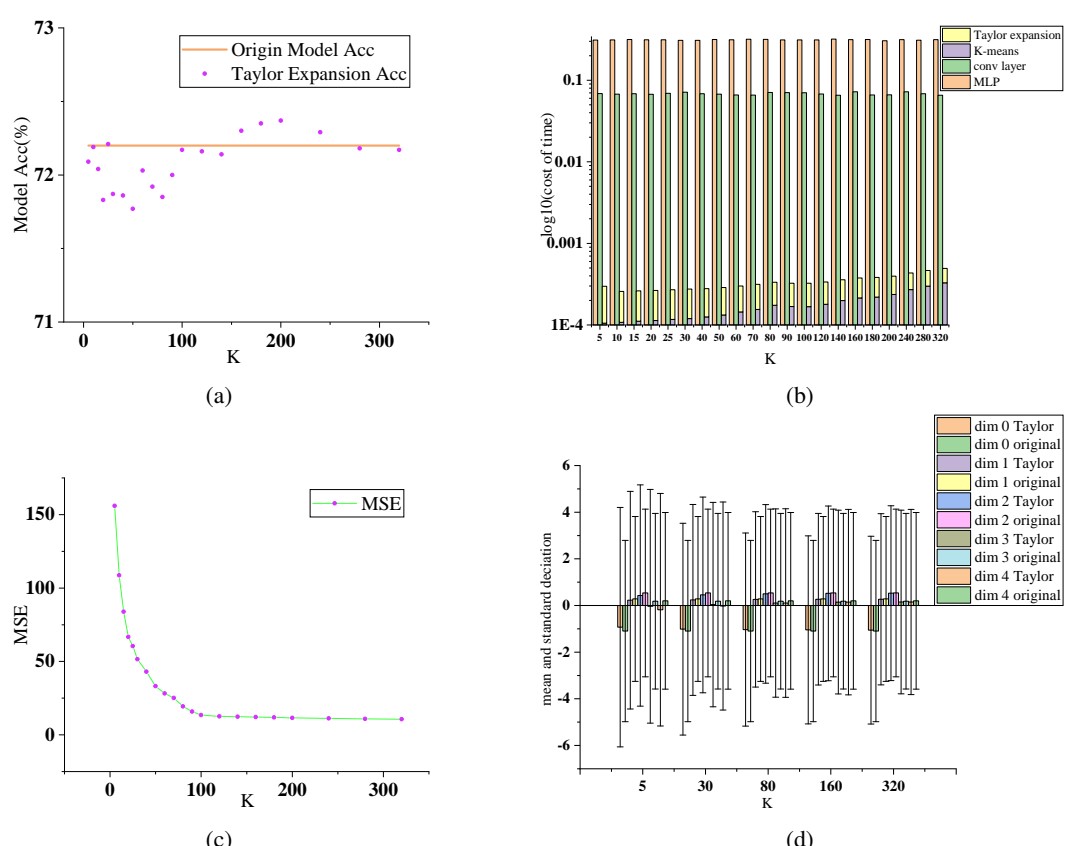

Figure 3: Experiments result of VGG-16 on Cifar-100. **(a)** : The comparison of Top-1 accuracy of original VGG-16 and its T-MLP on the Cifar-100 test set. **(b)** : The latency of the VGG-16 convolutional layers, MLP, and its T-MLP, including K-means prediction and Taylor expansion computation on Raspberry Pi 5. **(c)** : The mean square error between the 100-dimensional class-score vectors produced by the original VGG-16 model and T-MLP. **(d)** : The mean and standard deviation of the first five dimensions of the 100-dimensional output vectors generated by the original VGG-16 model and its T-MLP.

On this 100-class classification task, the T-MLP's top-1 accuracy retains the original top-1 accuracy to within $1\%$. And in terms of latency, far beyond our expectation, the T-MLP achieves a surprising $> 100\times$ inference acceleration. When the convolutional backbone is included, end-to-end inference is still accelerated by $6\times$. To quantify fidelity, we measured the Mean-Square Error(MSE) between the two output distributions. MSE decreases monotonically with increasing $k$, indicating progressively tighter approximation fidelity. Inspection of the first five dimensions of output vectors

corroborates this trend: at $k = 5$ the standard deviations differ by $> 1$, whereas at $k = 320$ both means and variances are statistically indistinguishable.

## 4.2 NETWORK INTRUSION DETECTION

Table 2: MLP Experiments on Raspberry Pi 5. ACC denote the accuracy of T-MLP. $T_{MLP}$, $T_{kmeans}$, $T_{expansion}$ respectively denote the latency of the original MLP, the K-means prediction, and the Taylor expansion computation. And $T_{T-MLP}$ reports the latency of our proposed method

| K | Acc | MSE | $T_{MLP}$ | $T_{kmeans}$ | $T_{expansion}$ | $T_{MLP}$ | $T_{T-MLP}$ |
|---|---|---|---|---|---|---|---|
| 5 | 85.13 | 43.10 | 5.67E-4 | 8.19E-5 | 7.02E-5 | 5.67E-4 | 1.52E-4 |
| 20 | 96.60 | 6.84 | 5.26E-4 | 6.20E-5 | 4.90E-5 | 5.26E-4 | 1.11E-4 |
| 35 | 96.73 | 3.89 | 5.79E-4 | 6.30E-5 | 5.10E-5 | 5.79E-4 | 1.14E-4 |
| 50 | 96.97 | 3.09 | 5.28E-4 | 6.26E-5 | 4.96E-5 | 5.28E-4 | 1.12E-4 |
| 90 | 97.75 | 1.38 | 5.06E-4 | 6.57E-5 | 4.98E-5 | 5.06E-4 | 1.16E-4 |
| 150 | 98.04 | 0.90 | 5.14E-4 | 6.81E-5 | 5.06E-5 | 5.14E-4 | 1.18E-4 |

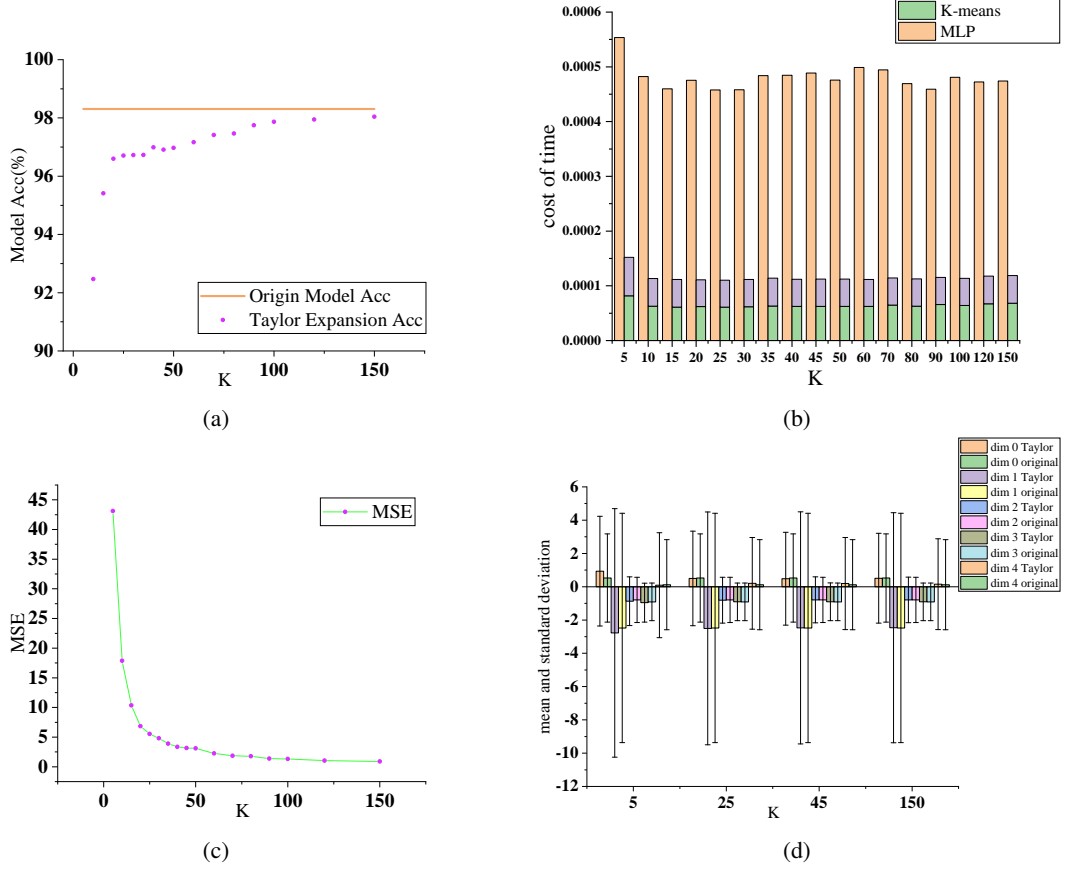

(a)

(b)

(c)

(d)

Figure 4: Experiments result of MLP on NSL-KDD. **(a)** : The comparison of Top-1 accuracy of original MLP and its T-MLP on the NSL-KDD test set. **(b)** : The latency of the MLP, and its T-MLP, including K-means prediction and Taylor expansion computation on Raspberry Pi 5. **(c)** : The mean square error between the 23-dimensional class-score vectors produced by the original MLP and its T-MLP. **(d)** : The mean and standard deviation of the first five dimensions of the 23-dimensional output vectors generated by the original MLP and its T-MLP.

To secure resource-constrained edge networks, real-time identification of anomalous traffic is essential. Network-intrusion detection is thus a critical task at the edge-side network. We evaluate our

method on NSL-KDD(Tavallaee et al., 2009), a de-facto standard benchmark for intrusion-detection systems (IDS). For the network intrusion detection task, we trained a three layer MLP on the NSL-KDD dataset. The MLP layers are dimensioned as (42, 256), (256,96), and (96, 23), respectively, and we use ReLU as activation functions. The model achieves 98.31% top-1 accuracy on the NSL-KDD test set. Analogous to the preceding experiment, we replace the three-layer MLP with T-MLP, while progressively increasing the $k$ values from 5 to 150. The experimental results are depicted in Table 2 and Figure 4.

In contrast to the previous experiment, the Taylor expansion counterpart on NSL-KDD starts with a pronounced accuracy deficit for small $k$, yet the gap narrows rapidly and collapses to 0.3 % when $k$ reaches 150. In terms of latency, the T-MLP still achieves a $4\times$ inference-speedup. And the counterpart reproduces the previously observed fidelity trend: MSE versus the original MLP decreases monotonically with the increment of $k$, and the empirical mean and standard deviation of the first-five dimensions of output vectors converge to their original values.

## 4.3 ABLATION STUDIES

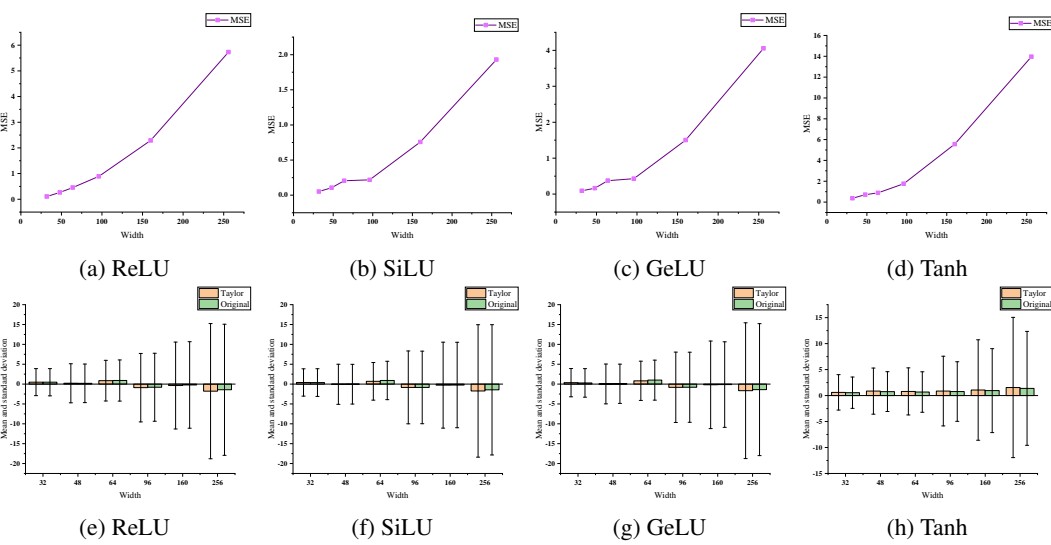

|  (a) ReLU | (b) SiLU | (c) GeLU | (d) Tanh |
| (e) ReLU | (f) SiLU | (g) GeLU | (h) Tanh |

Figure 5: This figure shows how the model's output evolves with increasing width after the MLP is replaced by its T-MLP. **(a-d)** : how the mean-square error between the outputs of the original MLP and its T-MLP varies as the model's width increases for ReLU, SiLU, GeLU and Tanh activation functions respectively. **(e-h)** : how the mean and standard deviation between the outputs of the original MLP and its T-MLP varies as the model's width increases for ReLU, SiLU, GeLU and Tanh activation functions respectively.

To systematically investigate how the architectural complexity of MLPs influences the fidelity of their Taylor expansion counterparts, we conducted a controlled ablation study. We utilized the canonical breast cancer dataset(Wolberg & Street, 1993) to train multi-layer perceptrons (MLPs) of varying widths and heights. After randomly reserving 20 % of the instances as a held-out test set, we fitted a K-means model with k = 32 on the remaining training data.

Architecture sweeps were performed along two independent axes: width and height. In the width ablation experiment, height was held constant at 3 layers, while the hidden width increased from 32 to 256. And in height ablation experiment, width was fixed at 32 per hidden layer, while the number of hidden layers progressively increased. For every configuration we recorded three summary statistics: MSE, mean, and standard deviation between the outputs of the original MLP and its corresponding T-MLP. Additionally, we assessed the sensitivity of our approximation to the choice of activation function by repeating the above protocol under four canonical activations: ReLU, SiLU, GeLU and Tanh. The input, hidden, and output layers were dimensioned as (30, width), (width, width), and (width, 1), respectively. And all models are trained on the training set for 50 epochs. Comprehensive results are reported in Figure 5,6.

Figures 5 and 6 illustrate how the approximation capability of the T-MLP evolves with model width and height under four distinct activation functions. As the width and height of the original MLP increase, its expressive power grows accordingly, leading to a more complex and rapidly varying loss landscape. Consequently, the first-order Taylor expansion employed by T-MLP struggles to capture these fine-grained nonlinearities, and the approximation capacity consistently deteriorates regardless of the activation function used. This trend suggests that deeper and wider networks require a larger number of centroids to maintain acceptable fidelity.

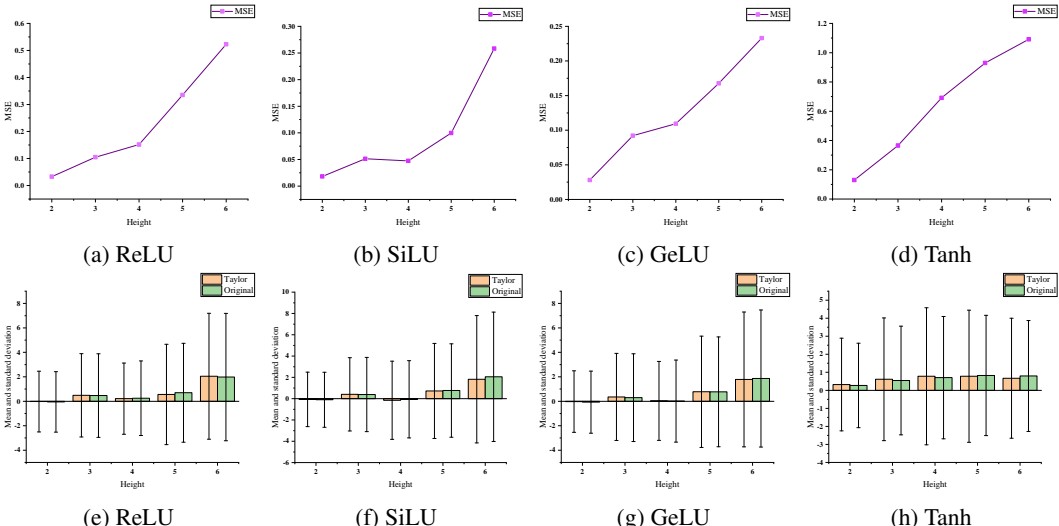

| (a) ReLU | (b) SiLU | (c) GeLU | (d) Tanh |
|---|---|---|---|

| (e) ReLU | (f) SiLU | (g) GeLU | (h) Tanh |
|---|---|---|---|

Figure 6: This figure shows how the model's output evolves with increasing height after the MLP is replaced by its T-MLP. **(a-d)** : how the mean-square error between the outputs of the original MLP and its T-MLP varies as the model's height increases for ReLU, SiLU, GeLU and Tanh activation functions respectively. **(e-h)** : how the mean and standard deviation between the outputs of the original MLP and its T-MLP varies as the model's height increases for ReLU, SiLU, GeLU and Tanh activation functions respectively.

We also evaluate the acceleration effect of T-MLP under varying model widths and heights, as illustrated in the Figure 7. The input, hidden, and output layers were dimensioned as (512, width), (width, width), and (width, 1), and k is set to 512. As predicted by our complexity analysis, the computational gap between the original MLP and T-MLP widens with height and width increasing. At height=3 and width=512, our method already delivers $> 5\times$ speed-up. Moreover, the acceleration grows super-linearly with increasing width and height, confirming that T-MLP is especially advantageous for large-scale T-MLP.

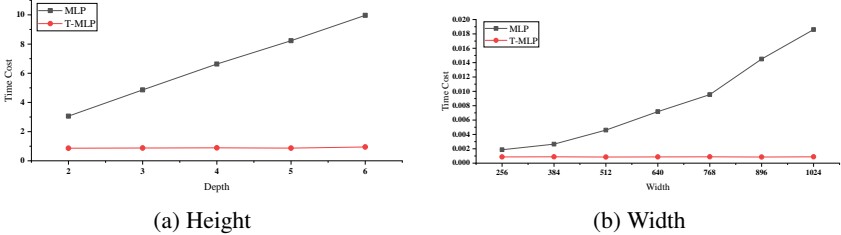

| (a) Height | (b) Width |
|---|---|

Figure 7: The influence of model height and width on speed-up

## 5 DISCUSSIONS

In this section, we interpret the observed phenomena and derive the conditions under which optimal acceleration–accuracy trade-offs are achieved.

To begin with, we examine how MLP scale conditions the inference acceleration of our approach. Experiments consistently reveal a positive scaling law: speed-up grows super-linearly with model scale. This trend is first guaranteed by the reduced theoretical complexity, and is further amplified by hardware properties.

As illustrated in Figure 8, although K-means prediction initially incurs slightly higher latency than matrix multiplication at small scales, the crossover occurs as dimensionality grows, and the gap widens thereafter. This is because, despite identical asymptotic complexity, K-means distance computation outperforms matrix multiplication in practice due to reduced memory write-backs and superior SIMD utilization.

On the other hand, the experimental platform itself also matters. On contemporary CPUs, GEMM performance transitions from a compute-bound to a memory-bandwidth-bound regime once the matrix scale exceeds the threshold of cache. And as illustrated in Figure 8, the latency of matrix multiplication does not exhibit a quadratic relationship with matrix size; instead, it undergoes a step-wise increase once the scale exceeds the cache threshold. Yet this overhead is mitigated by our approach, which drastically shrinks the matrix-multiplication footprint. The Raspberry Pi 5 is equipped with a 512 KB L1 cache and a 2MB L2 cache and VGG16's classifier exceeds 60 MB, which is far beyond the cache threshold. Whereas our method confines the matrix multiplication size to the L1 cache, yielding orders-of-magnitude speed-ups.

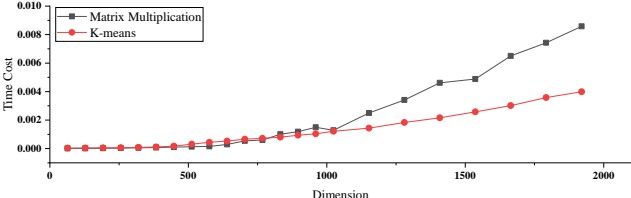

Figure 8: Time cost of different configurations of matrix multiplication $(1 \times N) \times (N \times N)$ and K-means prediction$(k = N, dim = N)$.

Next, we examine the fidelity of our approximation. The upper bound reveals that approximation fidelity hinges on the the local curvature (Hessian norm) and the distance between the actual input and its assigned centroid. The Euclidean distance between them monotonically decreases with the number of clusters k. Thus, a larger k yields a strictly tighter bound in expectation.

Empirically, the required K is strongly modulated by the data geometry. In Image Classification, VGG-16 delivers 512-D deep features that are highly coherent and clustering-friendly; consequently k = 320 already suffices to approximate the MLP. Conversely, in the Network Intrusion Detection experiment, the NSL-KDD dataset provides only 42 raw hand-crafted metrics whose scales, units, and marginal distributions are heterogeneous, thus a larger k is required. However, both theoretical complexity and empirical experiments indicate that the latency of T-MLP is less sensitive to the choice of k than to model width, validating the adoption of a larger k in practice.

In summary, our method excels when (i) the number of centroids k is sufficiently large, (ii) the MLP is of substantial scale, and (iii) the target device is memory-bandwidth-constrained.

## 6 CONCLUSION

We propose a novel approach for efficient MLP inference on edge devices by replacing the original network with its first-order Taylor expansion anchored at K-means centroids, achieving substantial latency reduction. Theoretical analysis and extensive experiments corroborate the effectiveness of the proposed method, and we further delineate the sufficient conditions for maintaining optimal performance.

In the future work, we will incorporate second- and higher-order Taylor expansions and systematically investigate the trade-offs among cluster budget k, expansion order, and inference latency.

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

## A  APPENDIX

In this section, we derive an upper bound on the approximation error for the vector-valued function $f : R^d \to R^m$ by means of the integral remainder.

Assume that $z_t = c_i + t(x - c_i), t \in [0, 1]$ denote the line segment between $x$ and $c_i$, and function f is twice differentiable along the line segment. The first order Taylor expansion is:

$$f(x) = f(c_i) + \nabla f(c_i)^T (x - c_i). \tag{5}$$

And the integral remainder is:

$$f(x) - \hat{f}(x) = \int_0^1 (1 - t)\nabla^2 f(z_t)[x - c_i, x - c_i]dt. \tag{6}$$

The Euclidean norm of the difference between the original value and its Taylor-expansion approximation is:

$$||f(x) - \hat{f}(x)|| \le \int_0^1 (1 - t)\nabla^2 f(z_t)[x - c_i, x - c_i]dt. \tag{7}$$

Since

$$||\nabla^2 f(z_t)[x - c_i, x - c_i]|| \le sup_{||u||=1}||\nabla^2 f(z_t)[u, u]|| \cdot ||x - c_i||^2 \tag{8}$$

Factor out the distance term and perform integration.

$$||f(x) - \hat{f}(x)|| \le ||x - c_i||^2 \int_0^1 (1 - t)\nabla^2 f(z_t)_{op}dt. \tag{9}$$

Extract the supremum of the Hessian norm over the line segment.

$$H^{(i)}_{max}(x) := sup_{\in[c_i,x]}||\nabla^2 f(z)||_{op} \tag{10}$$

$$\int_0^1 (1 - t)||\nabla^2 f(z_t)_{op}||dt \le H^{(i)}_{max}(x) \int_0^1 (1 - t)dt = \frac{1}{2}H^{(i)}_{max}(x) \tag{11}$$

Then we obtain the final upper bound.

$$||f(x) - \hat{f}(x)|| \le \frac{1}{2}sup_{z\in[c_i,x]}||\nabla^2 f(z)||_{op} \cdot ||x - c_i||^2 \tag{12}$$

