# OpenReview forum: "Replacing Multi-Layer Perceptron with Taylor Expansion"
_ICLR.cc/2026/Conference — ICLR 2026 Conference Withdrawn Submission_

### Official Review · Reviewer_wU3S · 2025-10-26

**Soundness:** 2
**Presentation:** 2
**Contribution:** 2
**Rating:** 2
**Confidence:** 4

**Summary:**

This paper proposes T-MLP, an approximation-based MLP acceleration method for inference on edge devices. T-MLP leverages first-order Taylor expansion to approximate MLP. To identify the expansion points of the Taylor expansion, it uses the cluster centers from $K$-means algorithm applied on the outputs of the layer before MLP. Experiments are done to validate the effectiveness of the proposed method.

**Strengths:**

1.The idea of using a simple Taylor expansion to approximate the result of MLP is interesting.

2.The paper is well-written and the structure is clearly arranged.

**Weaknesses:**

1. The technical contribution is limited. The derived error upper bound of T-MLP is essentially the error upper bound of a first-order Taylor expansion.

2. It is difficult to determine the $K$ value for $K$-means algorithm in practice. Different datasets may have different preferences.

3. The experiments are not comprehensive.

	- There is no comparison with other existing inference acceleration models. It is necessary to compare T-MLP with other acceleration methods under the same experimental settings, such as the methods based on model compression and Approximate Matrix Multiplication mentioned in the related work.

	- The datasets used are inconsistent across different experiments. The breast cancer dataset in the ablation studies does not appear in other experiments, which is confusing.

	- More experiments on large datasets are expected.

   - In Figure 8, what is the purpose of comparing time cost of matrix multiplication with K-means prediction instead of T-MLP as a whole?

	- In Table 1, the latency of T-MLP is not reported. There is no unit for the latency as well. According to the table, T-MLP achieves around 5.5 to 5.8 speedup. The 6x speedup described in the paper is not precise.

	- In Table 1, Table2, Figure 3 and Figure 4, why do the values of T_{conv}, T_{MLP}, T_{expansion}, and T_{conv+MLP} change as $K$ increases while they do not involve any $K$-means calculation?

**Questions:**

1. How about using higher-order Taylor expansions? What is the trade-off between accuracy and latency?

2. The ablation study should be experiments that involve replacement of Taylor expansion or $K$-means algorithm? The current ones are more like parameter studies.

---

### Official Review · Reviewer_1k6E · 2025-10-28

**Soundness:** 2
**Presentation:** 2
**Contribution:** 2
**Rating:** 4
**Confidence:** 4

**Summary:**

The paper proposes T-MLP, an inference-time approximation of a trained MLP using first-order Taylor expansion around a set of K-means centroids computed from inputs observed at the MLP layer during training. For each centroid ci, the method precomputes f(ci) and the Jacobian ∇f(ci), then approximates f(x) ≈ f(ci) + ∇f(ci)ᵀ(x−ci) where ci is the nearest centroid to x at inference. The claimed result is that an L-layer MLP is replaced by: (i) a nearest-centroid lookup (K-means prediction) and (ii) one matrix–vector multiply with the precomputed Jacobian, plus two vector additions. The authors present a time-complexity argument suggesting advantages for wide/deep MLPs with appropriate k, and provide an error bound based on the Hessian norm and ∥x−ci∥². Experiments on Raspberry Pi 5 show large speedups for the VGG-16 classifier on CIFAR-100 (>100× for the MLP module; ~6× end-to-end) with <1% top-1 accuracy drop, and ~4× for a 3-layer MLP on NSL-KDD.

**Strengths:**

- Simple idea with practical angle: local linearization of an MLP via first-order Taylor around K-means centroids; replaces multi-layer evaluation by one nearest-centroid dispatch and one matmul.
- Reasonable theoretical framing: time-complexity comparison; error bound scaling with Hessian norm and centroid distance; qualitative analysis of when it works (large/wide MLPs, memory-bound CPUs, larger k for better fidelity).
- Edge-device focus: measurements on Raspberry Pi 5 for two workloads; ablations over k, width/height, and activations provide some coverage of behavior.

**Weaknesses:**

- Memory/footprint is not addressed. For VGG-16’s classifier (4096→100), storing ∇f(ci) per centroid implies a Jacobian of size [4096×100] per ci. With k=320, that is ~41M floats ≈ 160 MB (plus f(ci)), which is not “hardware-friendly” on resource-constrained devices. The method can easily exceed available memory and cache—contradicting a central claim. A careful memory and storage analysis is needed, along with quantization/compression baselines.
- Timing anomalies and methodology. Table 1 suggests `Tconv+T−MLP` is sometimes lower than `Tconv` alone, which is not physically plausible; K-means prediction times (≈1e−4 s) for high-dimensional, k up to 320 on a Pi 5 also look overly optimistic. Provide: (i) methodology (warmup, repetitions, variance), (ii) isolation of cache/warm-start effects, and (iii) end-to-end CPU profiling. Include optimized BLAS baselines and vectorized nearest-centroid implementations for apples-to-apples.
- Limited generality and baselines. The classifier head of VGG-16 is a convenient case; show results on deeper/wider MLP blocks where output dimensionality is large (e.g., transformers’ intermediate projections), and compare to strong compression/approximation baselines (quantization + operator fusion; low-rank; structured pruning) on the same device. Also compare against product-quantization-based MLP approximation, which is a closer methodological baseline.
- Accuracy–k trade-off and OOD. Fidelity relies on k-means over in-distribution hidden features; how robust is dispatch for OOD inputs or non-stationary distributions? Report OOD tests and drift robustness. The ablations suggest larger width/height exacerbate error—how large must k grow to maintain accuracy? What is the resulting memory/time impact?
- Novelty claim. Approximating nonlinear mappings by local linear models around cluster centers is a well-trodden path; positioning relative to local linear/MoE approximations and AMM/PQ-inspired MLP approximations is underdeveloped. The contribution here is a specific instantiation for MLP acceleration; novelty is incremental without stronger empirical validation.

**Questions:**

- Memory/compression: What is the exact storage footprint (bytes) for f(ci) and ∇f(ci) across all k in each experiment? Can you quantize/compress Jacobians (e.g., 8-bit or low-rank) while retaining accuracy and speed? Provide a memory–latency–accuracy Pareto.
- Nearest-centroid cost: Report vectorized K-means dispatch cost as k and dimension grow; compare to a fused two-layer MLP matmul with optimized BLAS. Does the crossover point occur at the same thresholds on other CPUs/NPUs?
- Beyond classifier heads: Can you show results on larger output dimensions and deeper MLP stacks (e.g., replacing two or three FC layers in a transformer block) where the Jacobian is even larger? How do speed and memory scale there?
- Higher-order expansions: Second-order (or piecewise quadratic) approximations could reduce k; do you have preliminary results quantifying the trade-off vs. storage and compute?

---

### Official Review · Reviewer_3zkF · 2025-10-31

**Soundness:** 3
**Presentation:** 3
**Contribution:** 2
**Rating:** 4
**Confidence:** 4

**Summary:**

The paper highlights that MLPs are among the fundamental building blocks of deep learning models and are highly computationally and memory-intensive. Towards this, the paper presents a method for converting an MLP into a light-weight module that joins k-means with Taylor series expansion. The module has been demonstrated to be deployable on edge devices and shown to be highly effective in reducing computational cost and latency requirements.

**Strengths:**

1. The paper presents a novel light-weight method for approximating MLPs to reduce their computational complexity for deployment on edge devices.
2. The paper demonstrates that the proposed method can achieve a speedup of up to 100 times for large-scale MLPs.
3. The paper is well-written, well-structured, and easy to comprehend.

**Weaknesses:**

1. The experiments seem insufficient to demonstrate the effectiveness of the proposed technique, particularly compared with other compression techniques and across tasks of varying complexity. Additionally, the paper states that "our approach is orthogonal to existing methods and can be concatenated to yield simultaneously compact and highly efficient models for edge devices." This statement appears vague and unjustified, particularly given that the results of the proposed method vary significantly across model complexities in the ablation studies.
2. Only a limited set of models and datasets is considered for the experiments. The set is insufficient to draw any conclusions about which tasks, problems, and datasets the proposed method could be useful for, and for which it won't be effective. Additionally, the ablation studies are performed on an entirely different dataset without any specific justification.

**Questions:**

1. Is the proposed method really orthogonal to compression methods, especially pruning?
2. How does the effectiveness of the proposed technique vary with the complexity of the task (e.g., CIFAR10, CIFAR100, and ImageNet) and the relative complexity of the model, varied through pruning techniques?
3. Is the effectiveness of the proposed technique in the case of VGG-16 due to the model being too deep, and because the initial layers are handling a significant amount of classification workload, and the overall relative complexity of the model is too high compared to that of the problem?
4. What was the rationale behind the selection of VGG-16 for the experiment? If layer pruning, followed by neural/connection pruning, is applied to the model, will the gains remain the same?
5. Although Figure 1 provides a general idea of the motivation behind this work, it is not entirely clear. What is the core message the authors are trying to convey? How are the latency numbers computed?

---

### Official Review · Reviewer_x9Fm · 2025-11-03

**Soundness:** 2
**Presentation:** 4
**Contribution:** 2
**Rating:** 4
**Confidence:** 2

**Summary:**

To address the computational cost of NNs, the paper aims to replace the linear layers with a light-weight model by using the first order Taylor expansion. That way, multiple expensive dense GEMM becomes 1 K-means lookup, 1 dense matrix multiplication with precomputed gradient, and 2 element-wise adds.
The evaluation targets especially on the inference workload on resource constrained edge devices, showing 6x speedup, > 100x MLP speedup and < 1% accuracy loss for VGG-16/cifar-100.

**Strengths:**

1. Both image classification and network intrusion detection tasks are evaluated.
2. Ablation study is shown for scaling up with the model width.

**Weaknesses:**

1. As the author mentioned, the accurate Tylor expansion based approximation relies on tuning the set of centroids and precomputing the gradients. This offline precompute overhead is needed per model per dataset.
2. Since the linear layers of VGG-16 are only 3-layer MLP, I am not sure how the error could accumulate with more linear layers. Based on the ablation study about height ( number of layers ), this method is not so appealing for deep MLPs unless there is a way for scaling up & scaling out.

**Questions:**

1. Based on the ablation study, it might be directly usable for wide and deep MLP, which is the building block for SOTA language models. What is the plan for scaling up & scaling out?
2. What is the speed up when applying T-MLP on GPU?

---

### Note · Authors · 2025-11-18

**Comment:**

Thanks to all the reviewers' time and patience. Our work needs to be further refined to be published. After discussion among the coauthors, we decide to withdraw this paper. Sincerely thanks to the reviewers.

**Withdrawal Confirmation:**

I have read and agree with the venue's withdrawal policy on behalf of myself and my co-authors.